# Annual HIV screening rates for HIV-negative men who have sex with men in primary care

Courtney B. Spensley[1], Melissa Plegue[1,2], Robinson Seda[3], Diane M. Harper[1,4,5] *

**1** Department of Family Medicine, University of Michigan, Ann Arbor, MI, United States of America,
**2** Department of Biostatistics, University of Michigan, Ann Arbor, MI, United States of America, **3** Michigan Medicine, University of Michigan, Ann Arbor, MI, United States of America, **4** Department of Obstetrics and Gynecology, University of Michigan, Ann Arbor, MI, United States of America, **5** Department of Women's and Gender Studies, University of Michigan, Ann Arbor, MI, United States of America

* harperdi@med.umich.edu

**Data Availability Statement:** Data files are available from Michigan's Deep Blue database (https://doi.org/10.7302/00fw-qr09).

## Abstract

### Background

Men who have sex with men (MSM) account for most new HIV diagnoses in the US. Annual HIV testing is recommended for sexually active MSM if HIV status is negative or unknown. Our primary study aim was to determine annual HIV screening rates in primary care across multiple years for HIV-negative MSM to estimate compliance with guidelines. A secondary exploratory endpoint was to document rates for non-MSM in primary care.

### Methods

We conducted a three-year retrospective cohort study, analyzing data from electronic medical records of HIV-negative men aged 18 to 45 years in primary care at a large academic health system using inferential and logistic regression modeling.

### Results

Of 17,841 men, 730 (4.1%) indicated that they had a male partner during the study period. MSM were screened at higher rates annually than non-MSM (about 38% vs. 9%, p<0.001). Younger patients (p-value<0.001) and patients with an internal medicine primary care provider (p-value<0.001) were more likely to have an HIV test ordered in both groups. For all categories of race and self-reported illegal drug use, MSM patients had higher odds of HIV test orders than non-MSM patients. Race and drug use did not have a significant effect on HIV orders in the MSM group. Among non-MSM, Black patients had higher odds of being tested than both White and Asian patients regardless of drug use.

### Conclusions

While MSM are screened for HIV at higher rates than non-MSM, overall screening rates remain lower than desired, particularly for older patients and patients with a family medicine or pediatric PCP. Targeted interventions to improve HIV screening rates for MSM in primary care are discussed.

**Funding:** NCATS funding through UL1TR002240
The University of Michigan Rogel Cancer Center
P30CA046592.

**Competing interests:** The authors have declared
that no competing interests exist.

**Abbreviations:** MSM, men who have sex with men
(MSM); PCP, primary care provider (PCP); EMR,
electronic medical records (EMR); HIV, human
immunodeficiency virus (HIV); STI, sexually
transmitted infection (STI); CDC, Centers for
Disease Control (CDC); CMH, Cochran-Mantel-
Haenszel (CMH) test; GEE, Generalized Estimating
Equations (GEE) framework.

## Introduction

The incidence of HIV infection in the United States has decreased over the past decade, but
the number of new infections among men who have sex with men (MSM) has plateaued
around 26,000 per year [1]. In 2018, MSM accounted for 69% of all new HIV diagnoses [2].
The lifetime risk of HIV diagnosis among MSM is 88 times the risk of non-MSM [3].

Due to the higher risk of HIV infection among MSM, frequent screening is paramount to
prevention and timely treatment. Since 2006, the Centers for Disease Control (CDC) has rec-
ommended that sexually active MSM be screened for HIV at least once annually if HIV status
is negative or unknown. More frequent screening is recommended for MSM with higher risk
sexual behaviors [4]. As often as every 3 to 6 months, screening may be more effective in avert-
ing lifetime costs associated with HIV infection than annual screening [5].

Despite these recommendations, HIV screening remains suboptimal. The percentage of
undiagnosed infections among all MSM is estimated at 15% and is much higher among young
MSM [6]. Those with undiagnosed infection and those diagnosed but not receiving care are
responsible for most new HIV transmissions [7]. Although 77% of MSM surveyed for National
HIV Behavioral Surveillance self-reported being screened in the past year [8], a recent system-
atic review found that self-reported 12-month screening rates for MSM vary widely, from 40%
to 71% [9]. MSM are tested in clinical settings, but these also vary widely and include primary
care and specialty clinics, emergency departments, urgent care, hospitals, sexually transmitted
infection (STI) clinics, and drug treatment programs, among others. The proportion of tests
done by primary care providers (PCPs) is therefore unclear. In addition, among non-MSM
there is always a portion who are bi-sexual, have not realized their sexual orientation, or
describe their sexual orientation in other terms and would benefit from the annual HIV
screening guidelines.

Previous studies have found that HIV-negative MSM with a PCP are more likely to be
tested for HIV than other men [10, 11] and a majority of MSM report that their PCP is aware
of their sexual orientation [12]. This suggests that PCPs are ideally positioned to improve HIV
screening for MSM.

Our primary study objective was to determine medical-record verified annual HIV screen-
ing rates in primary care across multiple years for HIV-negative MSM to estimate compliance
with guidelines. A secondary exploratory endpoint was to document rates for non-MSM in
primary care.

## Materials and methods

We performed a retrospective chart review of HIV-negative men with PCPs at Michigan Med-
icine. Primary care specialties were defined as family medicine, internal medicine, medicine-
pediatrics, or pediatrics. Michigan Medicine primary care includes 16 clinic locations, serving
three counties in Southeast Michigan. Patients included in this study were all males with no
prior HIV diagnosis between 18 and 45 years who had at least one primary care encounter
between March 2016 and March 2019. We chose 2016 as the index year because the question
of the sexual partners' gender became coded data elements with the clinic contact. Eligible
patients were grouped based on reported sexual partners at the most recent disclosure. Indi-
viduals who reported having a male sexual partner or both a male and female partner were
included in the MSM group. The non-MSM group included individuals who reported only a
female partner or no partner. Patients who did not answer the question were excluded from
the study. The study proposal was submitted to the Institutional Review Boards of the Univer-
sity of Michigan Medical School and was exempted from ongoing IRB review
(HUM00155091). Individual consent was waived for this study.

Data were obtained from a computerized search of the Michigan Medicine electronic medical records (EMR) and included patient age, race, sexual partner gender, HIV test order dates, self-reported alcohol consumption, and self-reported illegal drug use, in addition to provider specialty. All HIV test orders were included for analysis, regardless of whether a result was available. For patients who did not have an HIV test ordered, we additionally searched encounter notes for the terms "HIV" and "STI" in conjunction with "counsel," "screen," and "test." The presence of these terms could indicate a pertinent discussion about testing for HIV. Annual rates of HIV screening were calculated in three 12-month time periods by counting at least one HIV test order for a unique person in that 12 -month time period divided by the identified MSM population in that same time frame. Additionally, rates of HIV discussion among MSM were similarly counted through medical note documentation for those without an HIV test order. The same procedure was completed separately for non-MSM. Chi-square tests, stratified by time period, compared HIV testing and discussion rates between MSM and non-MSM patients. A Cochran-Mantel-Haenszel (CMH) test was used to compare the difference in HIV testing and discussion rates between groups across the three time periods.

Associations with having an HIV test ordered were evaluated using a clustered logistic regression model under a Generalized Estimating Equations (GEE) framework to account for repeated measures on the same patient across time periods. Models were adjusted for age, race, substance use (alcohol and illegal drugs), provider specialty, and time period. The main covariate of interest was group (MSM vs non-MSM). Interactions between group and other variables were investigated using F-tests to evaluate their inclusion by jointly testing the significance of all parameters involved in the interaction term. To aid in the interpretation of significant interactions, marginal effects and probabilities at set levels of covariates were estimated. All statistics were performed using Stata 15.1 software.

## Results

In total, 17,841 men met the criteria for inclusion over all three years. About 40% had data in all three time periods, 30% had data in two time periods, and the remaining 30% had data in only a single time period. Records identified 730 (4.1%) patients as having a male partner at some point during the study period. Compared with the non-MSM group, MSM patients were younger, were more often Hispanic, and more likely to use alcohol and illegal drugs (Table 1).

Primary outcome: HIV test ordering. The HIV test ordering rates for MSM ranged between 35–43% per year, significantly higher than the 9% for non-MSM in each year (**Table 2**). The CMH test for homogeneity of odds ratios across the three years was not significant (p-value = 0.19), suggesting this difference in HIV test ordering rates between groups was consistent across the three time periods. For encounters that did not have an HIV test ordered, MSM encounter notes were significantly more likely to contain HIV terms than non-MSM notes, suggesting a screening discussion occurred (MSM 20% vs non-MSM 10% in each year, p-values<0.001), but, likewise, this was also not significantly different across the three years. In addition, the combined discussion of *any* HIV and/or STI terms in the encounter notes was not different between MSM and non-MSM over all three years CMH homogeneity test (p-value = 0.12).

The logistic regression model did not find a significant effect of time-period on the likelihood of ordering an HIV test (Table 3). However, when adjusting for group, age, PCP department, race, alcohol and illegal drug use, all were associated with HIV test ordering. Older individuals were significantly less likely to have an HIV test ordered (aOR (95% CI) = 0.937 (0.93, 0.94)), and individuals with a history of alcohol use were more likely to have a test ordered (aOR (95% CI) = 1.18 (1.06, 1.30)). Patients of pediatricians were less likely to have an

**Table 1. Characteristics of MSM and non-MSM.**

| | MSM | Non-MSM | p-value |
|---|---|---|---|
| | **(n = 730)** | **(n = 17,111)** | |
| **Age**, mean(SD) | 30.9 (6.8) | 34.7 (6.9) | <0.001 |
| **Age Group**, n(%) | | | <0.001 |
| 18–25 | 180 (24.7) | 2,095 (12.2) | |
| 26–30 | 203 (27.8) | 2,847 (16.6) | |
| 31–35 | 153 (21.0) | 3,719 (21.7) | |
| 36–40 | 117 (16.0) | 4,109 (24.0) | |
| 41–45 | 77 (10.6) | 4,341 (25.4) | |
| **Race**, n(%) | | | 0.67 |
| White | 563 (78.2) | 12,857 (76.3) | |
| Black | 64 (8.9) | 1,591 (9.4) | |
| Asian | 65 (9.0) | 1,624 (9.6) | |
| Other | 28 (3.9) | 773 (4.6) | |
| **Hispanic**, n(%) | 37 (5.1) | 760 (4.6) | 0.49 |
| **Female Partner ever** n(%) | 172 (23.6) | 16,618 (97.1) | <0.001 |
| **Alcohol Use**, n(%) | 527 (80.2) | 12,101 (76.2) | 0.018 |
| **Illegal Drugs**, n(%) | 112 (17.8) | 1,726 (11.3) | <0.001 |
| **IV Drugs**, n(%) | 1 (0.2) | 12 (0.07) | 0.49 |
| **Department,** n(%) | | | 0.07 |
| Family Medicine | 347 (47.5) | 8,501 (49.7) | |
| Internal Medicine | 341 (46.7) | 7,354 (43.0) | |
| Internal Medicine-Pediatrics | 34 (4.7) | 1,111 (6.5) | |
| Pediatrics | 8 (1.1) | 145 (0.9) | |

HIV test ordered compared with patients of family medicine physicians (aOR(95% CI) = 0.22 (0.12, 0.40)) and internal medicine patients were more likely than family medicine patients (aOR (95% CI) = 1.19 (1.10, 1.30)).

Significant group interactions were found with race (interaction F-test p-value = 0.01) and illegal drug use (interaction F-test p-value = 0.006). Within MSM patients, there were no significant differences in the likelihood of test orders based on levels of drug use or race. However, in the non-MSM group, there were both significant race and drug use effects. **Fig 1** illustrates interaction results through the predicted probability of HIV tests being ordered under different levels of group, race, and self-reported illegal drug use.

The MSM group had a higher probability of HIV testing being ordered, compared with non- MSM, regardless of race or illegal drug use. Within the non-MSM group, race differences

**Table 2. Yearly HIV test order rates and noted HIV discussion by MSM vs. non-MSM.**

| | HIV TESTS ORDERED % (95% CI) | | | HIV TERMS IN NOTE WITHOUT HIV TEST ORDER, % (95% CI) | | |
|---|---|---|---|---|---|---|
| | **MSM** | **Non-MSM** | **p-value** | **MSM** | **Non-MSM** | **p-value** |
| **YEAR 1** | N = 478 | N = 11,235 | | N = 309 | N = 10,275 | |
| 3/1/16-2/28/17 | 35.3 (31.2, 39.8) | 8.5 (8.0, 9.1) | <0.001 | 21.4 (17.1, 26.3) | 10.9 (10.3, 11.5) | <0.001 |
| **YEAR 2** | N = 533 | N = 11,887 | | N = 335 | N = 10,870 | |
| 3/1/17-2/28/18 | 37.1 (33.1, 41.3) | 8.6 (8.1, 9.1) | <0.001 | 19.7 (15.8, 24.3) | 10.4 (9.9, 11.0) | <0.001 |
| **YEAR 3** | N = 550 | N = 12,394 | | N = 316 | N = 11,277 | |
| 3/1/18-2/28/19 | 42.5 (38.5, 46.7) | 9.0 (8.5, 9.5) | <0.001 | 20.6 (16.5, 25.4) | 10.9 (10.4, 11.5) | <0.001 |

CMH across years for MSM was not significant for HIV tests ordered or for HIV terms in encounter notes indicating there was no difference in rates among years.

**Table 3. GEE model results on HIV test order.**

|  | Odds Ratio | 95% CI | p-value |
|---|---|---|---|
| **Age** | 0.94 | [0.93, 0.94] | 0.000 |
| **Department** (Ref = Family Medicine) |  |  |  |
| Internal Medicine | 1.19 | [1.10, 1.30] | 0.000 |
| Internal Medicine, Pediatrics | 1.06 | [0.89, 1.25] | 0.504 |
| Pedicatrics | 0.22 | [0.12, 0.40] | 0.000 |
| **Alcohol Use** (Ref = none) | 1.18 | [1.06, 1.30] | 0.001 |
| **Drug Use** (Ref = none) | 1.85 | [1.63, 2.10] | 0.000 |
| **Race** (Ref = White) |  |  |  |
| Black | 3.10 | [2.72, 3.54] | 0.000 |
| Asian | 0.97 | [0.81, 1.16] | 0.715 |
| Other Race | 1.34 | [1.08, 1.66] | 0.007 |
| **Group** (Ref = Non-MSM) |  |  |  |
| MSM | 6.72 | [5.74, 7.88] | 0.000 |
| **Time** (Ref = Year 1) | 1.00 |  |  |
| Year 2 | 1.00 | [0.92, 1.09] | 0.971 |
| Year 3 | 1.01 | [0.93, 1.10] | 0.791 |
| *Group*Drug Use Interaction Term* |  |  |  |
| **MSM*Drug Use** | 0.62 | [0.45, 0.85] | 0.003 |
| *Group*Race Interaction Terms* |  |  |  |
| **MSM*Black** | 0.45 | [0.29, 0.69] | 0.000 |
| **MSM*Asian** | 1.29 | [0.81, 2.07] | 0.282 |
| **MSM*Other Race** | 0.85 | [0.45, 1.63] | 0.629 |
| *Drug Use*Race Interaction Terms* |  |  |  |
| **Drug Use*Black** | 0.70 | [0.55, 0.90] | 0.006 |
| **Drug Use*Asian** | 0.74 | [0.39, 1.42] | 0.368 |
| **Drug Use*Other Race** | 1.40 | [0.86, 2.29] | 0.177 |

were found between Black patients and both White and Asian, with Black patients having a higher probability of test ordered regardless of drug use. In all non-MSM racial groups, except Asians, there were significantly higher rates of HIV test ordering among drug users than those who self-reported no illegal drug use.

## Discussion

We are the first to document medically verified HIV test ordering from a large electronic health record of patients receiving primary care that was not a short-term quality improvement project [13]. Our results of an HIV test order rate are between 35–43% for MSM, considerably lower than the CDC or USPSTF goal [1, 2, 8, 14, 15]. Others have shown that men who self-disclose MSM status are more likely to have HIV testing ordered than not [16–18]. On the other hand, among the non-MSM, we document a much higher frequency of HIV testing in our study at 9% compared with 2% in other populations of non-MSM [13].

In addition, by analyzing data over three years, we find that the HIV testing rate for MSM is consistently low, thus establishing a reliable baseline annual testing rate. These results hold even after adjusting for self-report alcohol or illegal drug use, prominent risk factors for HIV outside of sexual orientation. We are also the first to show that among those MSM without an HIV test order entered into the health system, only 20% receive HIV counseling as

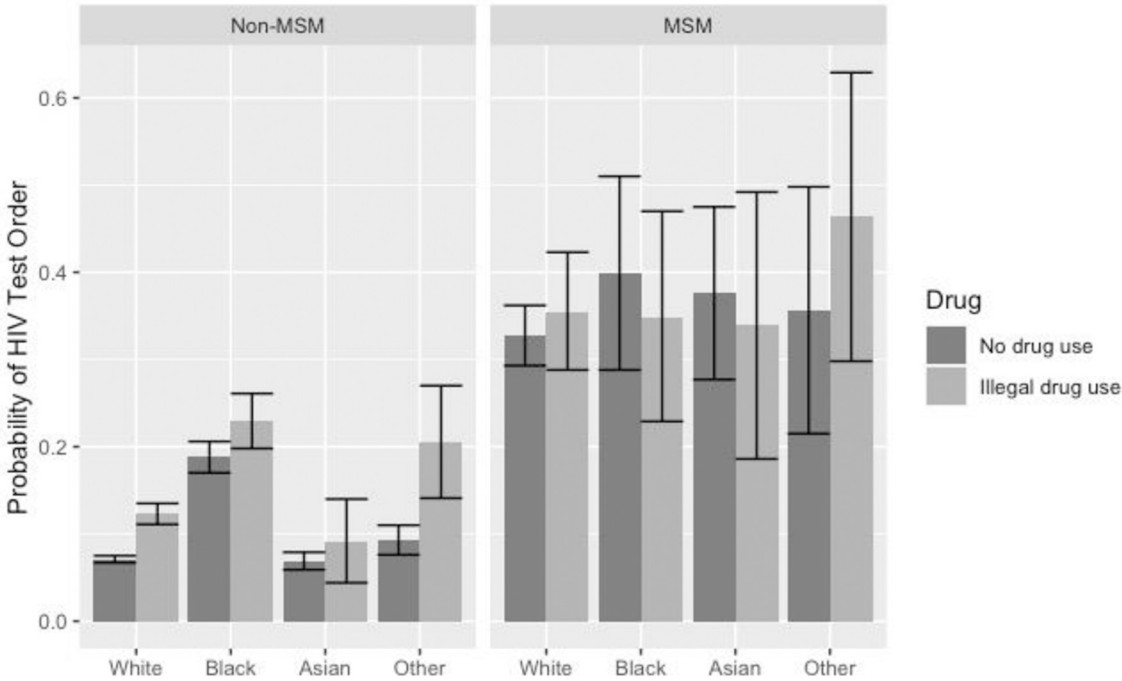

**Fig 1. Predicted probability of HIV test ordering by MSM v non-MSM, race, and Illegal drug use status.** The probabilities of HIV test ordering estimated from regression GEE modeling were adjusted for patient age, group (MSM vs non-MSM), race, PCP department, time, alcohol use and illegal drug use. Interactions of group*race, group*illegal drug use and race*illegal drug use were included, the group*race*drug use interaction was not significant.

documented in the text of the note, severely lower than is necessary to raise awareness and identify those who could benefit [4]. These baseline testing and counseling rates are essential for future quality improvement interventions to meet CDC or USPSTF screening goals.

Our results suggest that current systems are inadequate. We hypothesize that multi-level interventions with the health system, primary care physicians, and men themselves could be necessary to increase the HIV test ordering rate. The CDC estimates that 15% of men are unaware that they have HIV, in part, because they have never been tested [19]. The portion of MSM who do not have a PCP is unknown, as many seek care through STI clinics or Emergency Departments [20]. Even with a PCP, many MSM may not disclose their sexual orientation, often due to negative experiences or perceived quality of communication [21], despite recommendations to do so [22]. Studies about physician-related barriers to care reveal that some take a complete sexual history only when relevant to the chief complaint [23]. Even if the PCP is aware of the patients' sexual orientation, MSM-specific screening guidelines may not be known [24].

We showed that among non-MSM, HIV testing was greater among persons of color independent of self-reported alcohol and illegal drug use. These findings correlate to the successful increased public health campaigns to target Black and Hispanic men who are infected with HIV at disproportionately higher rates than White men. Despite this effort which has resulted in higher rates of HIV screening among Black and Hispanic men [11, 13, 25], no improved health outcomes have been documented from this increased testing [16]. Nevertheless, our work supports the need for universal screening to all races, potentially prompted by a best policy advisory from the EMR or national quality metrics [13, 26].

## Limitations

While the medically verifiable HIV test order is a strength of this work, the self-report characteristics of sexual practice, sexual orientation, alcohol use, and illegal drug use cannot be verified and may cause misclassification of risk factors and outcome populations [27]. As we stated in the methods section, patients with missing data for a sexual partner were not included in our analysis.

Moreover, we cannot verify the veracity of what was said in the clinic visit and what the PCP documented. While all patients who identified as MSM in our study had a male sexual partner by self-disclosure at some point during the study period, it is unclear whether the PCPS were aware of this designation before the office visit. Furthermore, while a proportion of encounter notes suggest discussion of HIV for MSM who did not have a test order, we cannot verify how accurately EHR documentation reflects actual patient-provider discussions.

Methodologically, our results are not weighted by population proportions across other demographic characteristics. Therefore, we cannot generalize results to a wider population than our clinic panels.

In addition, we are not able to evaluate the barriers physicians may have encountered in offering this testing. Barriers to HIV testing, such as lack of provider comfort with discussing sexual behavior and knowledge about HIV screening and treatment, are well documented and highlight problems with screening patients based on individual risk assessment [28] reinforcing the need for universal testing.

## Conclusion

While MSM are screened for HIV at higher rates than non-MSM, overall screening rates remain lower than desired, particularly for older patients and patients with a family medicine or pediatric PCP. Targeted interventions are needed to improve HIV screening rates for MSM in primary care.

## Acknowledgments

We thank the Data Office for Clinical & Translational Research at the University of Michigan Medical School and especially Chiu-Mei, Jeff Cowall, and Rino Srivastava for the coding required to extract EMR data from this project.

## Author Contributions

**Conceptualization:** Courtney B. Spensley, Diane M. Harper.

**Data curation:** Melissa Plegue, Robinson Seda.

**Formal analysis:** Melissa Plegue, Diane M. Harper.

**Funding acquisition:** Diane M. Harper.

**Methodology:** Courtney B. Spensley, Melissa Plegue, Robinson Seda, Diane M. Harper.

**Project administration:** Courtney B. Spensley, Robinson Seda.

**Resources:** Robinson Seda.

**Software:** Melissa Plegue.

**Supervision:** Diane M. Harper.

**Validation:** Robinson Seda, Diane M. Harper.

**Writing – original draft:** Courtney B. Spensley.

**Writing – review & editing:** Courtney B. Spensley, Melissa Plegue, Diane M. Harper.

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
