## [Decision Letter · Decision Letter 0]

9 Dec 2021

PONE-D-21-22682Annual HIV screening rates for HIV-negative men who have sex with men in primary carePLOS ONE

Dear Dr. Harper,

Thank you for submitting your manuscript to PLOS ONE. After careful consideration, we feel that it has merit but does not fully meet PLOS ONE’s publication criteria as it currently stands. Therefore, we invite you to submit a revised version of the manuscript that addresses the points raised during the review process.

The manuscript has been evaluated by two reviewers, and their comments are available below. The reviewers have raised a number of concerns that need attention. They request additional information on methodological aspects of the study, revisions to the statistical analyses and inclusion/exclusion criteria for patients. Could you please revise the manuscript to carefully address the concerns raised?

We look forward to receiving your revised manuscript.

Kind regards,

Elisa Panada

Associate Editor

PLOS ONE

Journal Requirements:

Reviewers' comments:

Reviewer's Responses to Questions

**Comments to the Author**

1. Is the manuscript technically sound, and do the data support the conclusions?

Reviewer #1: Partly

Reviewer #2: Yes

2. Has the statistical analysis been performed appropriately and rigorously? 

Reviewer #1: Yes

Reviewer #2: Yes

3. Have the authors made all data underlying the findings in their manuscript fully available?

Reviewer #1: Yes

Reviewer #2: No

4. Is the manuscript presented in an intelligible fashion and written in standard English?

Reviewer #1: Yes

Reviewer #2: Yes

5. Review Comments to the Author

Reviewer #1: Increasing rates of routine HIV screening in primary care practices is critical to ending the HIV epidemic. As such, the manuscript addresses an important issue. However, there are several points that I feel require clarification.

(1) Please provide a justification for limiting the sample to ages 18 to 45. The CDC recommendations are not limited to men in these age groups, so I see no reason not to include men of all ages. In fact, I would expect disparities in rates of testing by age to be even greater if older men had been included in the sample.

(2) Please also provide some discussion about how reliably data on sexual orientation are actually reported in the EHR. The authors do not mention any missing data on this variable, but I can't believe that there were no cases with sexual orientation excluded. Depending on the size of the sample, it may be necessary to run the analyses with three groups (MSM, non-MSM, and missing).

(3) Please reference the tables/figures in the narrative portion of the text to help guide the reader.

(4) I do not understand why the authors chose to examine the data across years. Is there any reason to expect an increase in HIV screening over the three-year period? I see no justification for that and it does not make sense to me. The question is about an individual's receipt of an HIV screening test, so I would have expected the analysis to focus at the individual level to see if any screening had occurred, controlling for the number of days/years the individual was included in the sample. Those with data across all three calendar years would have more opportunity to have a screening test than those who were in the sample for only one year. The finding of significant differences by year but no significance by the CMH test was confusing at first, because the authors provided no justification for expecting changes in screening patterns by year. The approach to analysis that I have suggested seems to me more in line with the stated objectives of the study.

(5) On a related note, in line 190, the authors note that the lack of an increasing in screening is "a disquieting finding." The low level of testing overall is certainly disquieting, but since the authors have not indicated a reason to expect an increase in annual rates of screening, this finding in and of itself is not especially "disquieting."

(6) Lines 165-166 appear to be missing something - the sentence is incomplete.

(7) The Limitations section should be expanded to include more discussion of the reliability of the methods used. Do the authors have any means to determine how regularly clinicians include in their notes discussions with patients about matters such as HIV testing, especially if the patient refused to be tested?

(8) Please provide details within the text regarding which characteristics the study's sample "matches the distribution in the state of Michigan." Is that only in regards to the percentage of men identified as MSM (4.1%) or is that across other demographic characteristics? If it is only in terms of the percentage, then the authors' claims of generalizability cannot be substantiated. It could be that the percentage is similar but that the study sample includes a higher percentage of white MSM than found in the state as a whole.

Reviewer #2: Thank you for allowing me to review your manuscript. Your research is clearly communicated and has meaningful implications to improve clinical practice. Please see a few comments/thoughts I had while reviewing.

on page 6, line 81, you mention non-MSM and state bi-sexual men. I understand those who may not be aware of their sexual identity, but was curious why you did not include bi-sexual men as MSM? Are you suggesting here that the bi-sexual men may not have engaged in a sexual encounter with another man or are you just directly placing bisexual men in a "non MSM" category?

Another question I had was the distribution of race/age based on practice setting. This is clearly not a focus of your work, but as you noted, STI clinics and ER's are frequently used sites; however, because you are the first to describe this data, did you see any associations with race/ethnicity and practice setting? Again, a question I had while reading that may have implications for clinical intervention implementation. And as a total side note, it would be interesting to see if the age/race/ethnicity/gender of the provider showed any significant outcomes.

it took me a few seconds to understand table 1, which made me wonder if it would make for a more clear presentation to have the M and SD in the column header. That is the format I think most readers are accustomed to seeing.

Finally, please consider changing your conclusions section to a Discussion, followed by your important conclusion statement. You have many important points to address and separating the discussion from your suggested conclusions may better frame the entirety of your discussion as well as making the clinical implications stand out to readers.

Thank you again for allowing me to review this manuscript. I enjoyed reading it and can clearly see the importance of this study.

6. PLOS authors have the option to publish the peer review history of their article (what does this mean?). If published, this will include your full peer review and any attached files.

Reviewer #1: No

Reviewer #2: No

---

## [Author Response · Author response to Decision Letter 0]

17 Jan 2022

January 17, 2022

Elisa Panada

Associate Editor

PLoS ONE

Re: PONE-#-21-22682: Annual HIV screening rates for HIV-negative men who have sex with men in primary care https://www.editorialmanager.com/pone/default1.aspx

Dear Dr Panada,

We are happy to provide the revisions to our manuscript as requested by the reviewers. 

Reviewer # 1 Response to reviewer Page

Please provide a justification for limiting the sample to ages 18 to 45. The CDC recommendations are not limited to men in these age groups, so I see no reason not to include men of all ages. In fact, I would expect disparities in rates of testing by age to be even greater if older men had been included in the sample. We thank you for this question. We agree that older men are equally important for HIV screening and would probably be even less screened than younger men, but we restricted our age range due to limited resources for data acquisition and analyst time. 

 -

Please also provide some discussion about how reliably data on sexual orientation are actually reported in the EHR. The authors do not mention any missing data on this variable, but I can't believe that there were no cases with sexual orientation excluded. Depending on the size of the sample, it may be necessary to run the analyses with three groups (MSM, non-MSM, and missing). Sexual orientation was determined based on the person’s response to an intake question asking them to disclose if they have a sexual partner. Data were extracted from the EHR only for patients who met the inclusion criteria of having a record of a sexual partner: male, female, none, or both. We cannot attest to the accuracy/reliability of self-disclosed sexual activity. 

We acknowledge this limitation in lines 219-xxx: “Patients with missing data for sexual partner were not included in our analysis.” 

 methods

Please reference the tables/figures in the narrative portion of the text to help guide the reader. The table/figure references are included in the narrative portion in bolded green font. 

I do not understand why the authors chose to examine the data across years. Is there any reason to expect an increase in HIV screening over the three-year period? I see no justification for that and it does not make sense to me The data were split into one year segments because the CDC recommendation is for eligible MSM to be screened at least once annually if his HIV status was negative or unknown. We used men with fixed encounter dates within 12 months to define the eligible population for screening in that year. We were able to extract the sexual orientation data for three distinct years. There was not a hypothesized change in screening rates over time; instead, we wanted to establish baseline annual screening rates, which we felt would be more accurate over a several-year period rather than a single year.

We presented the annual HIV screening rates for both MSM and non-MSM and found no difference year over year. This baseline is essential for future quality improvement projects to increase screening rates. We have added this in the discussion.

Lines 189-190: “In addition, by analyzing data over three years, we find that HIV testing rate for MSM are consistently low, thus establishing a reliable baseline annual testing rate.” 

Lines 195-xxx: “These baseline annual testing and counseling rates are important to establish for future quality improvement interventions to meet CDC or USPSTF screening goals.” 

The question is about an individual's receipt of an HIV screening test, so I would have expected the analysis to focus at the individual level to see if any screening had occurred, controlling for the number of days/years the individual was included in the sample. Those with data across all three calendar years would have more opportunity to have a screening test than those who were in the sample for only one year. Thank you. To clarify, analysis was performed at the person-year level. Individuals were included in each year period based on whether or not they had a record of any encounter during the time frame. If an individual did not have an encounter in a given year, they were not included in the denominator for screening rates, nor did they contribute to models for that year. Screening is meant to be done annually, and so individuals were included in each year as long as they had at least one encounter, regardless of prior screening status. 

The finding of significant differences by year but no significance by the CMH test was confusing at first, because the authors provided no justification for expecting changes in screening patterns by year. The approach to analysis that I have suggested seems to me more in line with the stated objectives of the study. We have stated the purpose of CMH testing in the Methods section in lines 112-114. However, we agree with the reviewer that the interpretation of the CMH test result was not clearly stated. We have revised the results section with the following, which more appropriately points out that the test finds consistent differences between the MSM and non-MSM ordering rates across the three time periods rather than suggesting we were looking for changes in rates across time.

Lines 137-xxx: “The CMH test for homogeneity of odds ratios across the three years was not significant (p-value=0.19), suggesting this difference in HIV test ordering rates between groups was consistent across the three time periods.” 

On a related note, in line 190, the authors note that the lack of an increasing in screening is "a disquieting finding." The low level of testing overall is certainly disquieting, but since the authors have not indicated a reason to expect an increase in annual rates of screening, this finding in and of itself is not especially "disquieting." We thank the reviewer and share his/her level of dissatisfaction with the wording. It is correct that we did not have reason to expect any change in rates of annual screening, as our intent was to establish baseline screening annual rates prior to initiating quality improvement interventions at our clinics. We are changing the wording to reflect that the rates did not increase over 3 years, but instead were consistent.

Lines 189-190: “In addition, by analyzing data over three years, we find that HIV testing rate for MSM are consistently low, thus establishing a reliable baseline annual testing rate.” 

Lines xxx-xxx (previously 165-166) appear to be missing something - the sentence is incomplete We used a green font to highlight the table and figures. It is apparent that the green color was not visible in the reviewer’s copy of the manuscript. The sentence is complete: “Figure 1 illustrates interaction results through the predicted probability of HIV test being ordered under different levels of group, race, and self-reported illegal drug use.” 

The Limitations section should be expanded to include more discussion of the reliability of the methods used. 

Do the authors have any means to determine how regularly clinicians include in their notes discussions with patients about matters such as HIV testing, especially if the patient refused to be tested? This is a retrospective review of medical records, and, as has been well-established, not all information is recorded in a clinic visit that a research visit would require. We include in our methods section our process for determining how clinicians include discussion with patients about HIV testing in their notes. 

Lines 104-106: “For patients who did not have an HIV test ordered, we additionally searched encounter notes for the terms “HIV” and “STI” in conjunction with “counsel”, “screen” and “test”. The presence of these terms could indicate a pertinent discussion about testing for HIV.”

Additionally, we acknowledge the limitations of documentation reliability in the limitations section beginning on line 220 and have added additional clarification as follows. 

Lines 223-xxx: “Furthermore, while a proportion of encounter notes suggest discussion of HIV for MSM who did not have a test order, we cannot verify how accurately EHR documentation reflects actual patient-provider discussions.”

Please provide details within the text regarding which characteristics the study's sample "matches the distribution in the state of Michigan." Is that only in regards to the percentage of men identified as MSM (4.1 %) or is that across other demographic characteristics? If it is only in terms of the percentage, then the authors' claims of generalizability cannot be substantiated. It could be that the percentage is similar but that the study sample includes a higher percentage of white MSM than found in the state as a whole. Our results are not weighted by population proportions which would have been the more accurate way to describe generalizability of our results to others in Michigan. Therefore, we have moved this paragraph to a limitation agreeing that we cannot state these are generalizable to a wider population other than our clinic panels. 

Lines 224-225: “Methodologically, while a strength of our work is that our population distribution of MSM and non-MSM matches the population distribution in the state of Michigan (MDHHS 2021), our results are not weighted by population proportions across other demographic characteristics. Therefore, we cannot generalize results to a wider population than our clinic panels.”

Reviewer # 2 

on page 6, line 81, you mention non-MSM and state bi-sexual men. I understand those who may not be aware of their sexual identity, but was curious why you did not include bi-sexual men as MSM? Are you suggesting here that the bisexual men may not have engaged in a sexual encounter with another man or are you just directly placing bisexual men in a "non MSM" category? To clarify, individuals were identified as MSM in our study if they had ever indicated having a male partner, regardless of whether or not they ever had a female partner. So, in that respect we did include bi-sexual men in our MSM group. We have clarified this in the methods section.

Lines 98-xxx: “Eligible patients were grouped based on reported sexual partners. Individuals who reported having a male sexual partner or both a male and female partner were included in the MSM group. The non-MSM group included individuals who reported only a female partner and those with neither male nor female partners.”

Another question I had was the distribution of race/age based on practice setting. This is clearly not a focus of your work, but as you noted, STI clinics and ER's are frequently used sites; however, because you are the first to describe this data, did you see any associations with race/ethnicity and practice setting? Again, a question I had while reading that may have implications for clinical intervention implementation. And as a total side note, it would be interesting to see if the age/race/ethnicity/gender of the provider showed any significant outcomes We did collect data on the practice setting and on the age/race/ethnicity/gender of both the providers and patients. We agree that this would be interesting for future analysis but is beyond the scope of this initial study. 

it took me a few seconds to understand table 1, which made me wonder if it would make for a more clear presentation to have the M and SD in the column header. That is the format I think most readers are accustomed to seeing. Yes, we understand that if all of the data were means and standard deviations, then the labeling would be most appropriate in the header. However, the bottom half of the table presents data in number and percent, therefore, we must keep the labeling as it is. 

Finally, please consider changing your conclusions section to a Discussion, followed by your important conclusion statement. You have many important points to address and separating the discussion from your suggested conclusions may better frame the entirety of your discussion as well as making the clinical implications stand out to readers. Thank you. We agree with this reviewer’s suggestion and have changed our conclusions section to discussion and added our conclusion statement beginning on line 231. 

Thank you again for allowing me to review this manuscript. I enjoyed reading it and can clearly see the importance of this study. Thank you for your kind words.

---

## [Decision Letter · Decision Letter 1]

28 Feb 2022

PONE-D-21-22682R1Annual HIV screening rates for HIV-negative men who have sex with men in primary carePLOS ONE

Dear Dr. Harper,

Thank you for submitting your manuscript to PLOS ONE. After careful consideration, we feel that it has merit but does not fully meet PLOS ONE’s publication criteria as it currently stands. Therefore, we invite you to submit a revised version of the manuscript that addresses the points raised during the review process.

ACADEMIC EDITOR:

Thank you for the responsive edits to some of the reviewer comments. However, I am requesting that you please address additional comments from Reviewer 1. With these clarifications, the paper will be reconsidered to see if it is acceptable for publication. I believe it will contribute to the important need for clinicians to do more regular HIV testing particularly for MSM and others at high ongoing risk for HIV acquisition (e.g., PWID)

We look forward to receiving your revised manuscript.

Kind regards,

Dawn K. Smith

Academic Editor

PLOS ONE

Reviewers' comments:

Reviewer's Responses to Questions

**Comments to the Author**

1. If the authors have adequately addressed your comments raised in a previous round of review and you feel that this manuscript is now acceptable for publication, you may indicate that here to bypass the “Comments to the Author” section, enter your conflict of interest statement in the “Confidential to Editor” section, and submit your "Accept" recommendation.

Reviewer #1: (No Response)

Reviewer #2: All comments have been addressed

2. Is the manuscript technically sound, and do the data support the conclusions?

Reviewer #1: No

Reviewer #2: Yes

3. Has the statistical analysis been performed appropriately and rigorously? 

Reviewer #1: I Don't Know

Reviewer #2: Yes

4. Have the authors made all data underlying the findings in their manuscript fully available?

Reviewer #1: No

Reviewer #2: No

5. Is the manuscript presented in an intelligible fashion and written in standard English?

Reviewer #1: No

Reviewer #2: Yes

6. Review Comments to the Author

Reviewer #1: The authors have addressed some of the questions I raised in my initial review, but a re-reading raised additional questions and some responses to earlier questions by myself and the other reviewer have not been adequately addressed.

(1) The authors have clarified the source of information regarding sexual orientation. However, there remains some confusion. Over what timeframe was the question regarding sexual partner asked? Over the past year? Lifetime? The way I have seen this question asked in a clinical setting is in regard to sexual preference (male, female, or both). Maybe that is what the authors intended to say (i.e., preference rather than "having a sexual partner" which implies that the question refers to whether or not the patient has a current sexual partner)? In any case, current sexual partner is not necessarily indicative of sexual orientation, and that should be mentioned in the Limitations section. In the Materials and Methods section, the authors state that the non-MSM group includes "those with neither male nor female partner." I'm assuming that the authors mean that the person indicated that they were not sexually active but the wording is awkward since it could be taken to mean that the patient reported sexual activity with a non-human partner. The discussion is also confusing because, in their response to reviewer comments and Discussion, the authors now state that "Patients with missing data for sexual partner were not included in our analysis." If the authors mean that a person is missing data on sexual activity, then it would be better to simply say that in the methods section. In any case, the authors should state the number of cases excluded due to missing data.

(2) How was the search of the encounter notes performed? Given the large number of encounters included in the study, I'm assuming this process was automated. The authors should provide more details regarding the methods of the search. The description is also unclear because the authors specifically reference "rates of HIV discussion among MSM." Table 2 suggests that the search was performed for all patients included in the study (i.e., including for non-MSM), but the description suggests otherwise. If the same procedure was followed for both groups (which I'm sure it was), then that should be stated clearly. Similarly, when describing the calculation of the annual testing rates, the authors only mention the calculation for MSM. For the sake of completeness, please indicate that the same procedure was completed separately for non-MSM.

(3) Table 1 as reformatted remains confusing as the table headers do not align with the columns and the p-value column appears to be missing. I disagree with the authors' statement to the second reviewer that "we must keep the labeling as it is." There are ways to improve the presentation of the data and that should be done. Tables and Figures should stand on their own, so all necessary clarifications should be made.

(4) In the first paragraph of the Results section, please take out the statement, "... consistent with state-specific data for generalizability." Per my earlier comments and the authors' acknowledgement, this statement is confusing and not valid in any case. Please also remove the reference to generalizability from the discussion section (i.e., the paragraph that starts, "Methodologically, while a strength...") or at least limit it the context of the clinical sites.

(5) In the Discussion section, the authors state, "Our results also indicate that multi-level interventions with the health system, primary care physicians, and men themselves could be necessary to increase the HIV test ordering rate." The study does not address the specific barriers to testing, so this statement (even if true) is not specifically substantiated by the study. The authors may justify this statement on the basis of other research, but not on the basis of the study as reported.

(6) Please provide the tables describing the logistic regression models. The discussion is difficult to follow without data to reference.

Reviewer #2: (No Response)

7. PLOS authors have the option to publish the peer review history of their article (what does this mean?). If published, this will include your full peer review and any attached files.

Reviewer #1: No

Reviewer #2: No

---

## [Author Response · Author response to Decision Letter 1]

19 Mar 2022

March 10, 2022

Dawn K Smith

Re: PONE-#-21-22682R1: Annual HIV screening rates for HIV-negative men who have sex with men in primary care

Dear Editor Smith,

Thank you for the opportunity to respond a second time to Reviewer #1. We apologize for any miscommunication between our responses and the understanding of our comments.

Reviewer Comment Response Page

The authors have clarified the source of information regarding sexual orientation. However, there remains some confusion. Over what timeframe was the question regarding sexual partner asked? Over the past year? Lifetime? The question that is asked of each person is “What is the sex of your partner?” and the boxes to check are one for female, one for male, and a comment box. This sexuality question is meant to be reviewed at every visit but is often answered only at the first visit. A person can check none, one, or both boxes each time the question is updated.

We have changed the sentence to say:

Eligible patients were grouped based on reported sexual partners at the most recent disclosure.

 5

The way I have seen this question asked in a clinical setting is in regard to sexual preference (male, female, or both). Maybe preference rather than "having a sexual partner" question refers to whether or not the patient has a current sexual partner)? In any case, a current sexual partner is not necessarily indicative of sexual orientation, and that should be mentioned in the Limitations section. 

 We do not intend for the response to the question of who one has sex with to indicate sexual orientation. We define MSM literally to be men who have sex with men, not an orientation. 

In the Materials and Methods section, the authors state that the non-MSM group includes "those with neither male nor female partner." I’m assuming that the authors mean that the person indicated that they were not sexually active but the wording is awkward since it could be taken to mean that the patient-reported sexual activity with a non-human partner. 

 Thank you for drawing this inappropriate attribution to our wording to our attention.

We have changed the sentence to read:

The non-MSM group included individuals who reported only a female partner or no partner. 5

The discussion is also confusing because, in their response to reviewer comments and Discussion, the authors now state that "Patients with missing data for a sexual partner were not included in our analysis." If the authors mean that a person is missing data on sexual activity, then it would be better to simply say that in the methods section. We have added this sentence to the methods section:

Patients who did not answer the question were excluded from the study. 5

The authors should state the number of cases excluded due to missing data. that is what the authors intended to say We apologize that this is still a confusing point in the manuscript and have attempted to clarify the text to make it more clear. We extracted data from the EHR on all eligible men who indicated being sexually active. If they were not sexually active, they would not have completed partner information and therefore not be included in the extraction. We included all men who were extracted and grouped them into either those who had some indication of a male partner vs. those who had only female or no known partner, but sexually active with humans.

How was the search of the encounter notes performed? Given the large number of encounters included in the study,

I'm assuming this process was automated. The authors should provide more details regarding the methods of the search.

 We appreciate the reviewer’s concern for accuracy. We state in the methods that the terms HIV counsel…, HIV screen… HIV test… as well as STI counsel… STI screen… and STI test were programmed in the automated search. We feel that this is clear and that providing more coding detail would be confusing to the reader with no additional value. 6

The description is also unclear because the authors specifically reference "rates of HIV discussion among MSM." 

Table 2 suggests that the search was performed for all patients included in the study (i.e., including for non-MSM), but the description suggests otherwise. If the same procedure was followed for both groups (which I'm sure it was), then that should be stated clearly. 

 As noted below, we have clarified that the same procedure was used for MSM and non-MSM by stating:

The same procedure was completed separately for non-MSM. 6

Similarly, when describing the calculation of the annual testing rates, the authors only mention the calculation for MSM. For the sake of completeness, please indicate that the same procedure was completed separately for non-MSM. Thank you. We have added this sentence:

The same procedure was completed separately for non-MSM. 6

Table 1 as reformatted remains confusing as the table headers do not align with the columns and the p-value column appears to be missing. I disagree with the authors' statement to the second reviewer that "we must keep the labeling as it is." There are ways to improve the presentation of the data and that should be done. Tables and Figures should stand on their own, so all necessary clarifications should be made We profusely apologize for Table 1. We agree that the table as presented to you was massively mis-formatted. We have corrected this issue. 7

In the first paragraph of the Results section, please take out the statement, "... consistent with state-specific data for generalizability." Per my earlier comments and the authors' acknowledgment, this statement is confusing and not valid in any case. Please also remove the reference to generalizability from the discussion section (i.e., the paragraph that starts,

"Methodologically, while a strength...") or at least limit it the context of the clinical sites We apologize. We have now removed this statement and its references from the results section and from the discussion section. 6

In the Discussion section, the authors state, "Our results also indicate that multi-level interventions with the health system, primary care physicians, and men themselves could be necessary to increase the HIV test ordering rate." 

The study does not address the specific barriers to testing, so this statement (even if true) is not specifically substantiated by the study. The authors may justify this statement on the basis of other research, but not on the basis of the study as reported. We thank you for noting this obscurity. We have changed the first two sentences of this paragraph to :

Our results suggest that current systems are inadequate. We hypothesize that multi-level interventions with the health system, primary care physicians, and men themselves could be necessary to increase the HIV test ordering rate. 10

Please provide the tables describing the logistic regression models. The discussion is difficult to follow without data to

reference. We have provided Table 3 which provides the details of the logistic regression model. 

We are grateful for your attention to detail and sincerely feel we have made all the changes and clarifications asked. 

Sincerely,

Diane M Harper MD MPH MS

Professor

University of Michigan

---

## [Editor Report · Decision Letter 2]

28 Mar 2022

Annual HIV screening rates for HIV-negative men who have sex with men in primary care

PONE-D-21-22682R2

Dear Dr. Harper,

We’re pleased to inform you that your manuscript has been judged scientifically suitable for publication and will be formally accepted for publication once it meets all outstanding technical requirements.

Kind regards,

Dawn K. Smith

Academic Editor

PLOS ONE

Additional Editor Comments (optional):

Thank you for clearly addressing the major comments from reviewers. The paper is much improved.
---

## [Editor Report · Acceptance letter]

6 Jul 2022

PONE-D-21-22682R2 

Annual HIV screening rates for HIV-negative men who have sex with men in primary care 

Dear Dr. Harper:

I'm pleased to inform you that your manuscript has been deemed suitable for publication in PLOS ONE. Congratulations! Your manuscript is now with our production department. 

Kind regards, 

on behalf of

Dr. Dawn K. Smith 

Academic Editor

PLOS ONE